# Neuroimaging Changes in the Sensorimotor Network and Visual Network in Bipolar Disorder and Their Relationship with Genetic Characteristics

**DOI:** 10.3390/biomedicines13040898

**Published:** 2025-04-08

**Authors:** Chunguo Zhang, Yiding Han, Haohao Yan, Yangpan Ou, Jiaquan Liang, Wei Huang, Xiaoling Li, Chaohua Tang, Jinbing Xu, Guojun Xie, Wenbin Guo

**Affiliations:** 1Department of Psychiatry, The Third People’s Hospital of Foshan, Foshan 528000, China; cgzhang1994@163.com (C.Z.); liangjiaquan@muc.edu.cn (J.L.); fshw2122680@126.com (W.H.); lixiaoling0632@163.com (X.L.); jimmytong1983@163.com (C.T.); fs_xujinbing@163.com (J.X.); 2Department of Psychiatry, National Clinical Research Center for Mental Disorders, and National Center for Mental Disorders, The Second Xiangya Hospital of Central South University, Changsha 410011, China; yidinghan1997@gmail.com (Y.H.); yanhaohao1995@gmail.com (H.Y.); ouyangpan33@csu.edu.cn (Y.O.)

**Keywords:** bipolar disorder, resting-state functional magnetic resonance imaging, sensorimotor network, visual network, gene expression

## Abstract

**Objective**: Patients with bipolar disorder (BD) may exhibit common and significant changes in brain activity across different networks. Our aim was to investigate the changes in functional connectivity (FC) within different brain networks in BD, as well as their neuroimaging homogeneity, heterogeneity, and genetic variation. **Methods**: In this study, we analyzed the seed points and whole-brain FC of the sensorimotor network (SMN) and visual network (VN) in 83 healthy controls (HCs) and 77 BD patients, along with their genetic neuroimaging associations. **Results**: The results showed that, compared to HCs, BD patients exhibited abnormal FC in the SMN and VN brain regions. However, after three months of treatment, there were no significant differences in SMN and VN FC in the brain regions of the patients compared to pre-treatment levels. Enrichment analysis indicated that genes associated with changes in FC were shared among different SMN seed points, but no shared genes were found among VN seed points. **Conclusions**: In conclusion, changes in SMN FC may serve as a potential neuroimaging marker in BD patients. Our genetic neuroimaging association analysis may help to comprehensively understand the molecular mechanisms underlying FC changes in BD patients.

## 1. Introduction

Bipolar disorder (BD) is a complex and severe chronic illness that significantly impairs psychosocial functioning and shortens life expectancy by approximately 10 to 20 years. It has a high heritability (around 70%) and shares genetic risk alleles with other psychiatric and medical conditions [1]. Additionally, individuals with BD have a higher suicide rate compared to those with other mental disorders, with a suicide risk about 20 to 30 times greater than that of the general population [2,3]. More than 1% of the global population is affected by BD, regardless of nationality or race [4]. BD adversely impacts social functioning, quality of life, and suicidal tendencies, particularly among young people [5,6]. The pathological physiology and genetic characteristics of BD have become increasingly focused on by researchers.

Functional magnetic resonance imaging (fMRI) is a revolutionary neuroimaging technique that is widely used in neuroscience and clinical medicine to explore emerging areas of biomarkers [7]. Resting-state fMRI, in particular, is notable for its independence from external stimuli, revealing spontaneous neuronal activity patterns through blood-oxygen-level-dependent (BOLD) signals [8]. This technique provides a unique perspective for the in-depth investigation of complex neural network connections and functional characteristics within the brain. Researchers are actively exploring how to utilize this information to understand the fundamental mechanisms of the nervous system and enhance strategies for disease diagnosis and treatment. Resting-state fMRI is frequently utilized to investigate functional connectivity (FC) in mental disorders like BD and schizophrenia [9,10]. Studies have shown that patients with BD exhibit decreased resting-state FC between the default mode network (DMN) and the salience network (SN), even though the sample included patients in various affective states, such as manic, depressive, mixed, and euthymic phases [11]. Furthermore, studies have shown that changes in SN FC can distinguish major depressive disorder, BD, and healthy controls [12]. Wang et al. discovered reduced variability in dynamic FC (dFC) between the posterior DMN and the right central executive network (CEN) in patients with BD [13]. Therefore, exploring the abnormalities in the FC states of BD patients may provide new insights into the brain pathology of BD.

Emotion, cognition, and motor functions can be linked to specific neural networks in the brain at rest. Naturally, both sensorimotor coordination and psychomotor performance are mediated through the sensorimotor circuitry [14]. The distinct regions of the brain within this system exhibit synchronized low-frequency (<0.1 Hz) activity variations, creating an extensive sensorimotor network (SMN) [15,16,17,18]. Within the SMN, thalamocortical circuits demonstrate reciprocal anatomical and functional linkages between sensorimotor cortices and thalamic nuclei [19]. The fMRI-derived FC analyses reveal that thalamocortical synchronization dynamically orchestrates the convergence of afferent sensory information and efferent motor commands during sensorimotor integration [20]. Studies have shown a significant relationship between interhemispheric FC in the SMN of BD subjects [21]. Patients with BD often exhibit abnormal brain connectivity in areas such as the control network (ECN), SN, and DMN. However, there is limited research on the SMN and VN in BD patients. Investigating these networks could enhance our understanding of the neurobiological basis of BD.

In studies that integrate neuroimaging with genetics, intermediate phenotypes hold potential as genetic markers linked to clinical outcomes [22]. To better understand the genetic architecture of BD, it is essential to gather phenotypic information beyond just the diagnosis, as relying solely on the diagnosis may not sufficiently correlate with any biological processes [23]. Evaluating genetically relevant clinical characteristics in BD patients can enhance the phenotypic and genetic homogeneity of the sample, thereby increasing its power, specificity, and comprehension of its genetic architecture [23,24]. However, the relationship between FC abnormalities in the SMN and VN and related genes in BD patients, compared to healthy controls (HCs), remains unknown.

In this study, we aimed to use resting-state fMRI to detect abnormal FC in the SMN and VN regions of the brains of BD patients compared to HCs, to enhance our understanding of the neurobiological basis of their clinical features. Additionally, we planned to follow up with patients three months after treatment to investigate changes in FC before and after treatment. Finally, our goal was to deepen our understanding of BD’s mechanisms by studying the relationship between FC abnormalities and genes in BD patients compared to HCs.

## 2. Materials and Methods

### 2.1. Participants

This study included 170 participants: 88 diagnosed with BD and 82 as HCs. The HCs were recruited from local communities and physical examination centers, while the BD patients were selected from the Department of Psychiatry at the Third People’s Hospital of Foshan.

The BD cohort’s inclusion criteria were as follows: (1) meeting the DSM-5 criteria for BD; (2) aged between 18 and 55 years; (3) right-hand dominance; (4) completed ≥6 years formal education; (5) undergoing pharmaceutical treatment.

The inclusion criteria for the healthy controls (HCs) were as follows: (1) absence of personal/familial psychiatric history; (2) age-matched with the patients (18–55 years); (3) right-handedness; (4) minimum sixth-grade education attainment.

The exclusion criteria applied to all participants comprised the following: (1) structural brain anomalies detectable via neuroimaging; (2) comorbid neuropsychiatric diagnoses; (3) substance-use disorders (including alcohol); (4) contraindications for magnetic resonance imaging; (5) diagnosed metabolic disorders; (6) current pregnancy or lactation status.

Psychological and cognitive assessments were performed using the scale. Exploratory eye movement and event-related potential detection were conducted simultaneously.

### 2.2. Procedure

Baseline protocol: All subjects underwent a 3.0 T brain MRI scan and provided clinical data: anthropometric (BMI), endocrine (TSH, FT3, FT4, cortisol), metabolic (TG, CHOL, HDL, LDL, FBG, uric acid), and cardiovascular (HR) parameters.

Longitudinal design: BD participants subsequently received standardized pharmacotherapy for 12 weeks. Post-intervention evaluation replicated the initial neuroimaging protocol and biochemical assessments during the post-therapeutic phase.

### 2.3. Imaging Data Acquisition and Preprocessing

Resting-state fMRI was acquired on a GE 3.0 T Signa Pioneer system employing the following sequence parameters: temporal resolution: TR/TE = 2000/30 ms; spatial sampling: 64 × 64 matrix with 24 cm FOV; volumetric coverage: 36 axial slices (4 mm thickness, no gap); sequence configuration: 90° flip angle; acquisition duration: 250 volumes (500 s scan time). The dubjects received standardized instructions for ocular closure, wakefulness maintenance, and physical immobilization. Scanner noise attenuation was achieved through noise-reduction earplugs, while motion-restriction foam padding minimized cephalic displacement during image acquisition.

Preprocessing was conducted using the Data Processing Assistant for Resting-State fMRI (DPARSF) 6.0 (http://rfmri.Org/DPABI, accessed on 4 April 2025) software in MATLAB2018b to enhance the data quality and reduce artifacts. Slice timing correction was applied to account for differences in acquisition time across slices, ensuring temporal consistency within each volume. Head motion correction was performed to minimize movement-related artifacts, with a maximum displacement threshold of 2 mm and an angular motion threshold of 2° [25]. The functional images were then spatially normalized to a standard MNI space with a 3 × 3 × 3 mm^3^ resolution to facilitate inter-subject comparisons. Spatial smoothing using an 8 mm full width at half-maximum Gaussian kernel was applied to enhance the signal-to-noise ratio while preserving functional specificity. To eliminate scanner-related signal drifts, linear trend removal was conducted, followed by band-pass filtering between 0.01 and 0.08 Hz to isolate low-frequency fluctuations associated with resting-state brain activity while reducing high-frequency noise and physiological artifacts [26]. These preprocessing steps were chosen to improve the data reliability, minimize motion-induced confounds, and optimize the detection of meaningful neural signals for subsequent analyses.

### 2.4. FC Analysis

We utilized seed-based interregional correlation for examining FC. This study investigated the FC between seed points of the SN and the DAN across the entire brain. Seed point coordinates were obtained from the CONN Toolbox [27]. After preprocessing, seed-based FC analysis was performed by calculating Pearson’s correlation coefficients between each voxel and seed ROIs across the brain, creating a comprehensive connectivity matrix. Fisher’s r-to-z transformation was then applied to individual correlation matrices to improve the normal distribution (see Appendix A for the seed point coordinates used in this study).

### 2.5. Gene Expression Data

Gene expression data were obtained from the Allen Human Brain Atlas (AHBA) [28], a comprehensive repository with gene expression information for nearly the entire human brain from six donors.

We mapped the expression levels of 15,633 genes from the AHBA onto the AAL116 template. These expression levels were then correlated with differential brain maps (T values) comparing patients and controls at baseline. To reduce spatial autocorrelation effects, we used Brain SMASH surrogate brain maps, identifying genes linked to FC changes. Subsequently, we conducted enrichment analysis. We performed spatial correlation analysis between the T values of FC differences and the gene expression values at specific seed points between patients and HCs at baseline (spatial correlation analysis of neuroimaging and genomics).

### 2.6. Statistical Analysis

Categorical variables underwent χ^2^ tests, while continuous measures were evaluated through independent-samples *t*-tests (SPSS 22.0) for intergroup contrasts between the patient and control cohorts. Statistical significance adhered to α = 0.05 thresholding.

Cross-sectional FC profiles were contrasted via two-sample *t*-tests comparing baseline patient–control dyads. Longitudinal within-subject contrasts employed paired *t*-tests to evaluate treatment-induced FC alterations. Gaussian random field (GRF) correction was applied to address multiple comparisons (voxel-wise *p* < 0.001; cluster-level *p* < 0.05), covarying for demographic (sex/age/education) and motion parameters. Post-treatment FC amplitude comparisons incorporated identical covariates with GRF-corrected post hoc evaluations.

## 3. Results

### 3.1. Demographic and Clinical Characteristics

This study initially included 88 HCs and 82 patients. Due to excessive head movement, five participants from each group were excluded. The final analysis, therefore, included 83 HCs and 77 patients. Furthermore, 38 patients completed the follow-up assessments. There were no significant differences in age, gender, or education level among these groups. For detailed information, refer to Appendix A.

### 3.2. The Treatment Outcome

Appendix A display the clinical characteristics of the 38 BD patients who underwent the follow-up assessments. By the endpoint, these patients exhibited significant clinical improvement compared to their baseline evaluations.

Additionally, Appendix A illustrate the correlations between demographic and clinical characteristics at baseline and before treatment. Here, HCs represents healthy controls, T1 BD denotes the 77 BD patients included in the study, T1 BD (follow-up) refers to BD patients who completed the follow-up before treatment, and T2 BD indicates BD patients after treatment.

### 3.3. FC Analysis in Pre-Treatment Patients with BD and HCs

At baseline, compared to the healthy control group, the lateral SMN (L) (left thalamus, right thalamus), lateral SMN (R) (bilateral thalamus), superior SMN (bilateral thalamus), medial VN (left thalamus, left supramarginal gyrus/IPG, right thalamus), and lateral VN (R) (bilateral thalamus) exhibited higher FC. In contrast, compared to the healthy control group, the lateral SMN (L) (right postcentral gyrus), lateral SMN (R) (right SOG/MOG, left lingual/fusiform gyrus, right fusiform/lingual gyrus, left postcentral gyrus), and lateral VN (L) (right insula/Rolandic operculum) exhibited lower FC. Table 1 and Figure 1 and Figure 2 provide more in-depth information.

### 3.4. FC Analysis in Pre-Treatment and Post-Treatment Patients with BD

Three months post-treatment, there were no significant differences in SMN and VN FC in the patients’ brain regions compared to the pre-treatment levels.

### 3.5. Correlation Analysis Results

The correlation analysis results revealed no significant associations between clinical data and patient FC.

### 3.6. Enrichment Analysis

The gene enrichment analysis (bubble plots) separately displays the enrichment results for positively and negatively correlated genes. Each seed point presents only one Spearman correlation plot, showing the result with the highest *p*-value (see Appendix A). Among these, ROI 4 has no associated genes. ROI 5 has only one associated gene, so a Spearman correlation plot is shown directly.

Genes related to changes in FC are shared among different SMN seed points (see Table 2), but there are no shared genes among VN seed points.

## 4. Discussion

Compared with HCs, individuals with BD demonstrate atypical FC in the SMN and VN brain regions. Nonetheless, following three months of treatment, no notable alterations were observed in the SMN and VN FC of the patients’ brain regions relative to pre-treatment levels. Transcriptome–neuroimaging correlation analysis indicated that certain genes may play a role in BD’s molecular mechanisms by modulating FC. Enrichment analysis revealed that genes linked to FC alterations are shared across different SMN seed points, whereas no common genes were identified among VN seed points. Clarifying the state of the affective disorder continuum, its defining biomarkers, and its potential classification as unitary psychosis holds substantial significance [29,30]. In summary, our findings enhance the comprehension of the relationship between microscale gene expression patterns and macroscale FC disruptions in BD, offering initial evidence for the neurobiological mechanisms underlying BD-related FC modifications.

The thalamus is composed of various anatomically unique nuclei, each of which participates in separate functional networks [31]. Studies have consistently shown that increased FC between the sensorimotor cortical areas and the thalamus is observed in bipolar patients, regardless of the illness phase [32,33,34]. Based on our research findings, the FC between both the SMN and VN and the thalamus was increased. This observation aligns with earlier evidence primarily showing increased FC between the thalamus and sensorimotor regions in BD patients considered as a group [32,33,34]. Thalamus–SMN connectivity profiles reveal intrinsic neural signaling dynamics within the thalamocortical sensorimotor circuitry, serving as a neurophysiological substrate for sensorimotor integration and behavioral modulation [19,35]. Regarding the VN–thalamus connection, evidence suggests that interocular interactions might occur in the LGN [36], either through local interthalamic circuits [37] or via the significant number of corticothalamic fibers providing feedback from striate and extrastriate visual regions [38,39]. Prior research using event-related potentials has demonstrated that individuals with BD experience deficits in both auditory and visual attention [40,41], which may be attributed to the aforementioned factors. The changes in FC between the thalamus and SMN or VN may serve as a potential diagnostic biomarker for BD patients and could provide some assistance in subsequent treatment.

In our study, BD patients exhibited reduced FC values in the SMN, involving brain regions such as the right postcentral gyrus, right SOG/MOG, left lingual/fusiform gyrus, right fusiform/lingual gyrus, and left postcentral gyrus. The postcentral gyrus, a notable gyrus situated in the lateral parietal lobe of the human brain, is part of the somatosensory cortex system. Research has shown that BD patients have less cortical thickness in the left postcentral gyrus compared to HCs [42,43]. Additionally, the postcentral gyrus constitutes the somatosensory cortex. Recent studies indicate that the somatosensory cortex significantly contributes to various stages of emotional processing, such as recognizing affective meaning in stimuli, generating emotional states, and regulating emotions [44]. The right SOG/MOG, left lingual/fusiform gyrus, and right fusiform/lingual gyrus are primarily involved in processing visual information. The right-side regions mainly handle visual input, while the left-side regions are associated with more advanced visual recognition tasks. People exhibiting abnormal FC in the SOG may experience anxiety more readily compared to those with typical FC in this region [45]. Studies have also found that BD patients exhibit lower regional homogeneity (ReHo) values in the right MOG compared to HCs [46]. The lingual gyrus significantly contributes to synthesizing visual data and internal cognitive signals. Pioneering research by Xu’s team initially documented diminished low-frequency fluctuation fractional amplitude in this brain region of BD patients [47]. Comparative analyses further revealed weakened rsFC in the fusiform gyrus of BD patients relative to HCs [48]. This suggests that the abnormal connectivity between the SMN and other brain regions in BD patients is crucial for understanding the pathophysiology of BD and also contributes to its diagnosis.

Additionally, in our study, BD patients exhibited increased FC between the VN and the left supramarginal gyrus/IPG, while showing decreased FC between the VN and the right insula/Rolandic operculum. The supramarginal gyrus, which structurally incorporates both somatosensory associative areas and mirror neuron systems, mediates multimodal sensorimotor coordination, coordinates action sequence formulation/kinetic realization, and underpins evolutionarily conserved language processing mechanisms [49,50]. Studies suggest that higher gray matter volume in the left supramarginal gyrus may be indicative of resilience to disease expression, rather than a sign of vulnerability, in those at high risk of BD [51]. Additionally, studies have shown that ReHo in the IPG is elevated in comparison to HCs [52]. The insular cortex plays a crucial role in emotional processing and cognitive control in BD. Prior research has identified a reduction in bilateral insular volume among BD patients, potentially linked to atypical emotional regulation in BD [53,54]. The right Rolandic operculum has been shown to play a role in emotional regulation [55]. Prior research on BD has indicated disrupted right-hemisphere connectivity, specifically involving the Rolandic operculum [56]. The currently observed abnormal changes in FC within the VN may provide additional evidence supporting the enhanced function of the visual system in patients with BD. At the same time, this provides a new approach for diagnosing BD.

The SMN encompasses functional regions such as the primary motor cortex, premotor cortex, cingulate cortex, supplementary motor area, and sensory cortices located in the parietal lobe [57]. This network has functional connections with other brain systems related to sensorimotor skills, including visual and auditory subsystems, enabling the recognition of external stimuli and guiding cognitive growth from the early stages of life [58,59]. In our study, we found abnormal FC between the SMN and vision-related brain regions. However, there was no significant correlation between the SMN and VN. Additionally, prior research has shown that there is no causal connection between the SMN and VN in BD patients [60]. Our gene enrichment analysis results also indicated that there are no shared genes between the SMN and VN. Therefore, we assumed that, in BD patients, the SMN and VN each have their own distinct FC abnormalities. Notably, after three months of treatment, no significant changes were observed in the SMN and VN FC of the patients’ brain regions compared to pre-treatment levels. Based on this finding, we speculated that FC alterations in the SMN and VN are more likely associated with the pathogenesis of BD, rather than being directly driven by the treatment mechanism.

Using the AHBA microarray dataset, we identified genes related to the spatial patterns of different FC in the SMN and VN in BD. After genetic neuroimaging association analysis, we found shared genes among different ROIs in the SMN, whereas no shared genes were found in the VN. The latest genome-wide association study (GWAS) in adults has uncovered 64 single-nucleotide polymorphisms (SNPs) significantly linked to BD risk, reinforcing the polygenic nature of BD’s risk structure [61]. Research on BD indicates that CACNA1C and ankyrin-G protein (ANK3) are each linked with distinct structural and functional neuroimaging traits [62,63]. In addition, two independent genome-wide association studies have identified loci that show significant associations with BD, neurotransmitter transporters (e.g., GRIN2A), proteins involved in signal transduction (e.g., DGKH), encompassing genes related to ion channels (e.g., CACNA1C and SCN2A), and synaptic plasticity proteins (e.g., ANK3) [64,65]. Genetic associations frequently overlap, and individual variants can be linked to multiple traits; similarly, various genetic variants may correspond to the same BD characteristic [66]. Recent research has connected the Enhancing Neuro Imaging Genetics through Meta-Analysis (ENIGMA) consortium’s standardized brain MRI phenotypes to underlying neurobiological mechanisms and genetic influences [67]. One study found that brain differences observed in case–control comparisons across ENIGMA psychiatric working groups, including ENIGMA-BD, partially reflected established genetic correlations between disorders [68]. Another study demonstrated that a shared pattern of cell-specific gene expression could help explain common cortical thickness alterations across various psychiatric conditions [69]. Research suggests that genetic influences and disease-related deficits impact gray matter in partially distinct areas of the mainly heteromodal association cortex, indicating combined effects contributing to symptom severity and cognitive dysfunction in the disorder [70]. Large-scale studies integrating neuroimaging and genomic data help explore the role of brain regions in bridging genetic risk and the behavioral manifestations of BD [67]. Notably, our gene neuroimaging analysis shows promising potential in linking the neuroimaging heterogeneity of the SMN and VN with gene expression, which may offer new insights into analyzing different networks in BD.

Several limitations of this study should be acknowledged. First, most of the BD patients received prolonged pharmacological treatment, which may have influenced their FC values and cognitive outcomes. Second, no significant differences were observed in SMN and VN FC after three months of treatment compared to baseline data, suggesting that a longer treatment duration may be required to achieve noticeable effects. Third, our study did not stratify the participants based on the clinical staging of BD. Fourth, the sample size for neuroimaging genetic analysis remained relatively small.

## 5. Conclusions

In conclusion, compared to HCs, BD patients exhibit abnormal FC in the SMN and VN brain regions. Additionally, gene enrichment analysis showed that genes associated with changes in FC are shared between different SMN seed points, but not between VN seed points. This finding suggests heterogeneity in brain changes and gene expression in BD, which may offer new insights into the network mechanisms of BD. At the same time, this provides a new approach for diagnosing BD.

## Figures and Tables

**Figure 1 biomedicines-13-00898-f001:**
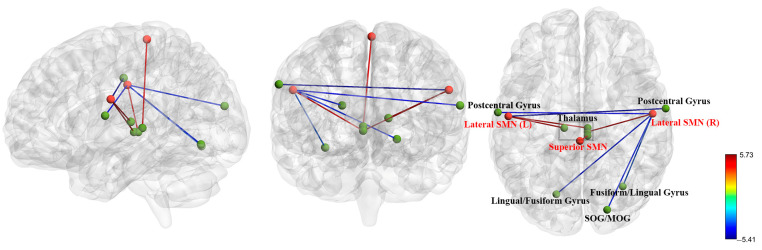
Functional connectivity abnormalities of the sensorimotor network were observed in BD patients at baseline compared to healthy controls. Red balls represent regions of interest, whereas green balls represent the brain regions connected to them. SMN = sensorimotor network; SOG = superior occipital gyrus; MOG = middle occipital gyrus; L = Left; R = right.

**Figure 2 biomedicines-13-00898-f002:**
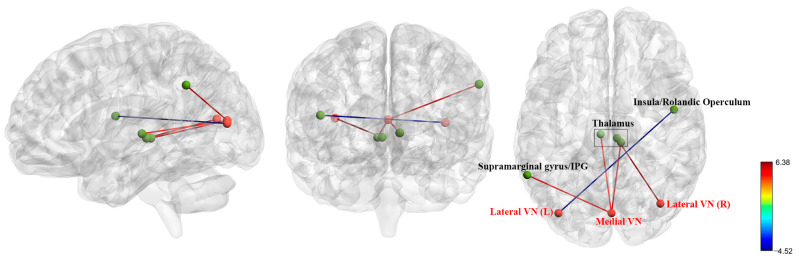
Functional connectivity abnormalities of the visual network were observed in BD patients at baseline compared to healthy controls. Red balls represent regions of interest, whereas green balls represent the brain regions connected to them. VN = visual network; IPG = inferior parietal gyrus; L = Left; R = right.

**Table 1 biomedicines-13-00898-t001:** The changes in functional connectivity within the sensorimotor network and visual network of BD patients at baseline compared to healthy controls.

ROIs	Brain Regions	MNI (x, y, z)	T Values	Cluster Size
Sensorimotor Network (SMN)				
Lateral SMN (L)	L Thalamus	−12, −21, 9	5.73	184
	R Thalamus	6, −21, 0	5.73	127
	R Postcentral Gyrus	66, −6, 33	−5.41	65
Lateral SMN (R)	Bilateral Thalamus	6, −24, 0	5.15	271
	R SOG/MOG	21, −84, 18	−3.95	167
	L Lingual/Fusiform Gyrus	−18, −72, −6	−3.94	116
	R Fusiform/Lingual Gyrus	33, −66, −12	−3.40	92
	L Postcentral Gyrus	−63, −9, 18	−4.09	80
Superior SMN	Bilateral Thalamus	6, −27, 3	4.82	135
Visual Network (VN)				
Medial VN	L Thalamus	−6, −21, 3	4.84	112
	L Supramarginal Gyrus/IPG	−60, −51, 36	5.19	102
	R Thalamus	9, −27, 0	5.65	100
Occipital VN	/	/	/	/
Lateral VN (L)	R Insula/Rolandic Operculum	48, −3, 15	−4.52	164
Lateral VN (R)	Bilateral Thalamus	6, −24, 0	6.38	423

ROI = region of interest; MNI = Montreal Neurological Institute; R = right; L = left; SOG = superior occipital gyrus; MOG = middle occipital gyrus; IPG = inferior parietal gyrus.

**Table 2 biomedicines-13-00898-t002:** The common and distinct FC alteration-related genes among ROIs.

ROIs	The Names of Common Genes That Showed Positive Correlations with FC Alterations	The Name of Common Genes that Showed Negative Correlations with FC Alterations	The Proportion of Common Genes in FC Alteration-Related Genes
Sensorimotor Network			
ROI 1&2	RFLNB	CMBL; EPHA5-AS1	1/1 in ROI 1; 1/10 in ROI 2
			2/3 in ROI 1; 2/24 in ROI 2
ROI 1&3	RFLNB	CMBL; EPHA5-AS1	1/1 in ROI 1; 1/12 in ROI 3
			2/3 in ROI 1; 2/22 in ROI 3
ROI 2&3	RFLNB	CCDC181; CMBL; EPHA5-AS1; HIST1H1A; LAMA3; MBP; PXDN; ZMYM2	1/10 in ROI 2; 1/12 in ROI 3
			8/24 in ROI 2 8/22 in ROI 3

FC = functional connectivity; ROI = region of interest; ROI 1 = lateral sensorimotor network (L); ROI 2 = lateral sensorimotor network (R); ROI 3 = superior sensorimotor network; R = right; L = left.

## Data Availability

The datasets utilized and/or examined in this study can be obtained from the corresponding author (Guo W) upon reasonable request.

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
