# Peer review of "Neuroimaging Changes in the Sensorimotor Network and Visual Network in Bipolar Disorder and Their Relationship with Genetic Characteristics"

_biomedicines, 2025, doi:10.3390/biomedicines13040898_

Round 1
Reviewer 1 Report
Comments and Suggestions for Authors
The manuscript presents valuable insights into neuroimaging changes in bipolar disorder, but several major issues need to be addressed:
A. Methodological Clarity – The inclusion and exclusion criteria, particularly regarding medication use, need to be better defined. Additionally, the justification for certain neuroimaging preprocessing steps should be clarified.
B. Post-Treatment Findings – The lack of significant changes after treatment raises concerns. A deeper discussion on whether the follow-up period was sufficient or if treatment effectiveness played a role is necessary.
C. Genetic Analysis Interpretation – The genetic findings are interesting but underdeveloped. A stronger connection between gene expression data and functional connectivity changes should be provided, along with comparisons to existing literature.
D. Discussion and Clinical Relevance – The results should be more explicitly linked to their clinical implications. How do these findings contribute to the diagnosis or treatment of bipolar disorder?
E. Structural and Linguistic Improvements – Some sections are redundant, while others lack depth. Improving logical flow and reducing overly technical language would enhance readability and clarity.
Addressing these issues will significantly strengthen the manuscript.
Author Response
Reviewer 1
The manuscript presents valuable insights into neuroimaging changes in bipolar disorder, but several major issues need to be addressed:
A. Methodological Clarity – The inclusion and exclusion criteria, particularly regarding medication use, need to be better defined. Additionally, the justification for certain neuroimaging preprocessing steps should be clarified.
Response: Thank you for your suggestion. We have refined the inclusion criteria for BD patients based on actual conditions and provided a more detailed explanation of the neuroimaging preprocessing steps. The specific details are as follows:
2.1 Participants
This study included 170 participants: 88 diagnosed with BD and 82 as HCs. HCs were recruited from local communities and physical examination centers, while BD patients were selected from the Department of Psychiatry at the Third People's Hospital of Foshan.
BD cohort inclusion: (1) meeting DSM-5 criteria for bipolar disorder; (2) aged between 18 and 55 years; (3) right-hand dominance; (4) completed ≥6 years formal education; (5) undergoing pharmaceutical treatment.
Healthy controls (HC): (1) absence of personal/familial psychiatric history; (2) age-matched with the patients (18-55 years); (3) right-handedness; (4) minimum sixth-grade education attainment.
Exclusion criteria applied to all participants comprised: (1) structural brain anomalies detectable via neuroimaging; (2) comorbid neuropsychiatric diagnoses; (3) substance use disorders (including alcohol); (4) contraindications for magnetic resonance imaging; (5) diagnosed metabolic disorders; (6) current pregnancy or lactation status.
Psychological and cognitive assessments were performed using the scale. Exploratory eye movement and event-related potential detection were conducted simultaneously.
2.3. Imaging data acquisition and preprocessing
Resting-state fMRI was acquired on a GE 3.0 T Signa Pioneer system employing the following sequence parameters: Temporal resolution: TR/TE = 2000/30 ms; Spatial sampling: 64 × 64 matrix with 24 cm FOV; Volumetric coverage: 36 axial slices (4 mm thickness, no gap); Sequence configuration: 90° flip angle; Acquisition duration: 250 volumes (500 s scan time). Subjects received standardized instructions for ocular closure, wakefulness maintenance, and physical immobilization. Scanner noise attenuation was achieved through noise-reduction earplugs, while motion-restriction foam padding minimized cephalic displacement during image acquisition.
Preprocessing was conducted using the Data Processing Assistant for Resting-State fMRI (DPARSF) software in MATLAB to enhance data quality and reduce artifacts. Slice timing correction was applied to account for differences in acquisition time across slices, ensuring temporal consistency within each volume. Head motion correction was performed to minimize movement-related artifacts, with a maximum displacement threshold of 2 mm and an angular motion threshold of 2°[25]. Functional images were then spatially normalized to a standard MNI space with a 3 × 3 × 3 mm³ resolution to facilitate inter-subject comparisons. Spatial smoothing using an 8 mm full-width at half-maximum Gaussian kernel was applied to enhance the signal-to-noise ratio while preserving functional specificity. To eliminate scanner-related signal drifts, linear trend removal was conducted, followed by band-pass filtering between 0.01 and 0.08 Hz to isolate low-frequency fluctuations associated with resting-state brain activity while reducing high-frequency noise and physiological artifacts [26]. These preprocessing steps were chosen to improve data reliability, minimize motion-induced confounds, and optimize the detection of meaningful neural signals for subsequent analyses.
B. Post-Treatment Findings – The lack of significant changes after treatment raises concerns. A deeper discussion on whether the follow-up period was sufficient or if treatment effectiveness played a role is necessary.
Response: Thank you for your suggestion! We have added a discussion on the FC changes before and after treatment. Additionally, since BD treatment is a long-term process, and the follow-up period in our study was only three months, which is relatively short, this may have contributed to the lack of significant differences on FC changes before and after treatment. This is indeed a limitation of our study, and we have addressed this issue in the limitations section. The specific description is as follows:
The SMN encompasses functional regions such as the primary motor cortex, premotor cortex, cingulate cortex, supplementary motor area, and sensory cortices located in the parietal lobe [57]. This network has functional connections with other brain systems related to sensorimotor skills, including visual and auditory subsystems, enabling the recognition of external stimuli and guiding cognitive growth from early life stages [58, 59]. In our study, we found abnormal FC between SMN and visually related brain regions. However, there was no significant correlation between SMN and VN. Additionally, prior research has shown that there is no causal connection between SMN and VN in BD patients [60]. Our gene enrichment analysis results also indicated that there are no shared genes between SMN and VN. Therefore, we supposed that in BD patients, SMN and VN each have their own distinct FC abnormalities. Notably, after three months of treatment, no significant changes were observed in the SMN and VN FC of patients' brain regions compared to pre-treatment levels. Based on this finding, we speculated that FC alterations in the SMN and VN are more likely associated with the pathogenesis of BD rather than being directly driven by the treatment mechanism.
Several limitations should be acknowledged. First, most BD patients received prolonged pharmacological treatment, which may have influenced FC values and cognitive outcomes. Second, no significant differences were observed in SMN and VN FC after three months of treatment compared to baseline data, suggesting that a longer treatment duration may be required to achieve noticeable effects. Third, our study did not stratify participants based on the clinical staging of BD. Fourth, the sample size for neuroimaging-genetic analysis remains relatively small.
C. Genetic Analysis Interpretation – The genetic findings are interesting but underdeveloped. A stronger connection between gene expression data and functional connectivity changes should be provided, along with comparisons to existing literature.
Response: Thank you for your suggestion. We have conducted a more in-depth discussion of the genetic analysis section to strengthen the association between the data and functional connectivity changes. The detailed description is as follows:
Using the AHBA microarray dataset, we identified genes related to the spatial patterns of different FC in SMN and VN in BD. After genetic neuroimaging association analysis, we found shared genes among different ROIs in the SMN, whereas no shared genes were found in the VN. The latest genome-wide association study (GWAS) in adults has uncovered 64 single-nucleotide polymorphisms (SNPs) significantly linked to BD risk, reinforcing the polygenic nature of BD's risk structure [61]. Research on BD indicates that CACNA1C and ankyrin-G protein (ANK3) are each linked with distinct structural and functional neuroimaging traits [62, 63]. In addition, two independent genome-wide association studies have identified loci that show significant associations with BD, neurotransmitter transporters (e.g., GRIN2A), proteins involved in signal transduction (e.g., DGKH), encompassing genes related to ion channels (e.g., CACNA1C and SCN2A), and synaptic plasticity proteins (e.g., ANK3) [64, 65]. Genetic associations frequently overlap, and individual variants can be linked to multiple traits; similarly, various genetic variants may correspond to the same BD characteristic [66]. Recent research has connected the Enhancing Neuro Imaging Genetics through Meta-Analysis (ENIGMA) consortium's standardized brain MRI phenotypes to underlying neurobiological mechanisms and genetic influences [67]. One study found that brain differences observed in case-control comparisons across ENIGMA psychiatric working groups, including ENIGMA-BD, partially reflected established genetic correlations between disorders [68]. Another study demonstrated that a shared pattern of cell-specific gene expression could help explain common cortical thickness alterations across various psychiatric conditions [69]. Research suggests that genetic influences and disease-related deficits impact gray matter in partly distinct areas of mainly heteromodal association cortex, indicating combined effects contributing to symptom severity and cognitive dysfunction of the disorder [70]. Large-scale studies integrating neuroimaging and genomic data help explore the role of brain regions in bridging genetic risk and the behavioral manifestations of BD [67]. Notably, our gene-neuroimaging analysis shows promising potential in linking the neuroimaging heterogeneity of SMN and VN with gene expression, which may offer new insights into analyzing different networks in BD.
D. Discussion and Clinical Relevance – The results should be more explicitly linked to their clinical implications. How do these findings contribute to the diagnosis or treatment of bipolar disorder?
Response: Thank you for your reminder. Changes in FC in BD patients play a significant role in diagnosis and treatment, such as serving as potential biomarkers. We have elaborated on this in the discussion section, and the specific details can be found in the manuscript's discussion part.
E. Structural and Linguistic Improvements – Some sections are redundant, while others lack depth. Improving logical flow and reducing overly technical language would enhance readability and clarity.
Response: Thank you for your suggestion. We have optimized the structure and language of the manuscript to improve logical flow and reduce overly technical terms, enhancing readability and clarity.
Addressing these issues will significantly strengthen the manuscript.
Reviewer 2 Report
Comments and Suggestions for Authors
This is reasonably good field study to investigate the biological mechanisms and fingerprints of bipolar disorder (BD). There is employed case-control design. The purpose of the study is to determine neural networks, belonging to sensorimotor and visual network and respective genetic alterations in patients with BD, as compared to healthy population. The methods are clearly described to allow replication. However the toolbox used in the data analysis is CONN, not COON. This has to be corrected. The introduction may be complemented with other research of immediate relevance, such as, but not limited to: https://doi.org/10.3390/biomedicines11061608
In the discussion, results should be interpreted in the same contexts, with a particular focus on the issue of definition of the affective spectrum [e.g. https://doi.org/10.1016/j.jad.2024.11.078; https://doi.org/10.1016/j.jad.2024.05.099]
The identification of neuroimaging and/or genetic biomarkers for BD is associated with the critical question whether there exist discrete boundaries between BD and other mood disorders. Therefore the lack of group of patients with MDD is likely to undermine the specificity of the reported findings and should be further stated as limitation.
Author Response
Reviewer 2
This is reasonably good field study to investigate the biological mechanisms and fingerprints of bipolar disorder (BD). There is employed case-control design. The purpose of the study is to determine neural networks, belonging to sensorimotor and visual network and respective genetic alterations in patients with BD, as compared to healthy population. The methods are clearly described to allow replication. However, the toolbox used in the data analysis is CONN, not COON. This has to be corrected. The introduction may be complemented with other research of immediate relevance, such as, but not limited to: https://doi.org/10.3390/biomedicines11061608
Response: Thank you for your suggestion. We have corrected "COON" to "CONN" and added new references in the introduction to further improve the manuscript.
In the discussion, results should be interpreted in the same contexts, with a particular focus on the issue of definition of the affective spectrum [e.g. https://doi.org/10.1016/j.jad.2024.11.078; https://doi.org/10.1016/j.jad.2024.05.099]
Response: Thank you for your suggestion. We have added an explanation of the results in the discussion section, with a particular focus on the definition of the affective spectrum, ensuring consistency within the same context. The detailed description is as follows:
Compared with HCs, individuals with BD demonstrate atypical FC in the SMN and VN brain regions. Nonetheless, following three months of treatment, no notable alterations were observed in the SMN and VN FC of patients' brain regions relative to pre-treatment levels. Transcriptome-neuroimaging correlation analysis indicates that certain genes may play a role in BD's molecular mechanisms by modulating FC. Enrichment analysis reveals that genes linked to functional connectivity alterations are shared across different SMN seed points, whereas no common genes are identified among VN seed points. Clarifying the state of the affective disorder continuum, its defining biomarkers, and its potential classification as unitary psychosis holds substantial significance [29,30]. In summary, our findings enhance the comprehension of the relationship between microscale gene expression patterns and macroscale FC disruptions in BD, offering initial evidence for the neurobiological mechanisms underlying BD-related FC modifications.
The identification of neuroimaging and/or genetic biomarkers for BD is associated with the critical question whether there exist discrete boundaries between BD and other mood disorders. Therefore, the lack of group of patients with MDD is likely to undermine the specificity of the reported findings and should be further stated as limitation.
Response: Thank you for your suggestion. This is indeed a limitation of our study, and we have addressed it in the limitations section. The specific description is as follows:
Several limitations should be acknowledged. First, most BD patients received prolonged pharmacological treatment, which may have influenced FC values and cognitive outcomes. Second, no significant differences were observed in SMN and VN FC after three months of treatment compared to baseline data, suggesting that a longer treatment duration may be required to achieve noticeable effects. Third, our study did not stratify participants based on the clinical staging of BD. Fourth, the sample size for neuroimaging-genetic analysis remains relatively small.
Round 2
Reviewer 1 Report
Comments and Suggestions for Authors
Dear Authors,
Thank you for your thorough revisions. The improvements in methodological clarity, genetic analysis, and clinical relevance significantly enhance the manuscript’s quality.